

# Look out: an exploratory study assessing how gaze (eye angle and head angle) and gait speed are influenced by surface complexity

Nicholas D.A. Thomas[1,2], James D. Gardiner[2], Robin H. Crompton[2] and Rebecca Lawson[1]

[1] Institute of Population Health Sciences, University of Liverpool, Liverpool, United Kingdom
[2] Institute of Ageing & Chronic Disease, University of Liverpool, Liverpool, United Kingdom

## ABSTRACT

**Background**. Most research investigating the connection between walking and visual behaviour has assessed only eye movements (not head orientation) in respect to locomotion over smooth surfaces in a laboratory. This is unlikely to reflect gaze changes found over the complex surfaces experienced in the real world, especially given that eye and head movements have rarely been assessed simultaneously.

**Research question**. How does gaze (eye and head) angle and gait speed change when walking over surfaces of different complexity?

**Methods**. In this exploratory study, we used a mobile eye tracker to monitor eye movements and inertia measurement unit sensors (IMUs) to measure head angle whilst subjects ($n = 11$) walked over surfaces with different complexities both indoors and outdoors. Gait speed was recorded from ankle IMUs.

**Results**. Overall, mean gaze angle was lowest over the most complex surface and this surface also elicited the slowest mean gait speed. The head contributed increasingly to the lowering of gaze with increased surface complexity. Less complex surfaces showed no significant difference between gaze and gait behaviour.

**Significance**. This study supports previous research showing that increased surface complexity is an important factor in determining gaze and gait behaviour. Moreover, it provides the novel finding that head movements provide important contributions to gaze location. Our future research aims are to further assess the role of the head in determining gaze location during locomotion across a greater range of complex surfaces to determine the key surface characteristics that influence gaze during gait.

# INTRODUCTION

Our ability to understand how people walk through their environment should be informed not only by assessment of their gait but by understanding the visual information available to them. Visual information is particularly important when environments are more complex, requiring increased planning to maintain stability whilst walking. For example, spatial and temporal visual information has been shown to be essential for correct foot positioning

Corresponding author
Nicholas D.A. Thomas,
N.Thomas3@liverpool.ac.uk

over complex surfaces, both inside and outside of the laboratory (*Matthis & Fajen, 2014*; *Matthis, Barton & Fajen, 2017*; *Matthis, Yates & Hayhoe, 2018*). Complex surfaces increase fall risk for all age groups as a result of poor stability (*Nyman et al., 2013*; *Talbot et al., 2005*). Therefore, an increased understanding of how vision and gait are impacted by different surfaces is important to help to understand and prevent falls.

Research investigating gait and gaze often uses terminology inconsistently. Here, we will use 'complex' to refer to all non-smooth surfaces. These include surfaces with slope changes (*Merryweather, Yoo & Bloswick, 2011*), inconsistently spaced foot targets (*Matthis & Fajen, 2014*; *Patla & Vickers, 2003*), uneven surfaces (*Thies, Richardson & Ashton-Miller, 2005*) and combinations of these features (*Marigold & Patla, 2007*; *Marigold & Patla, 2008*). Smooth surfaces here are taken to include even, horizontal surfaces in laboratories (*Marigold & Patla, 2007*), on walkways (*Graci, Elliott & Buckley, 2010*) and outside (*Storm, Buckley & Mazza, 2016*). Lastly, while gaze is often used by researchers to refer only to eye movements, here we define gaze as the orientation of the eye in a world reference frame. Gaze thus combines eye-in-head movements and head-in-world movements, which we measured using an eye tracker and an inertia measurement unit sensor (IMU) respectively.

Most gait research uses a smooth, horizontal, hard laboratory floor. However, some laboratory-based studies have started to address how conditions more representative of real-world surfaces may impact our behaviour. These studies have not, though, produced consistent findings. For example, *Menant et al. (2009)* found that gait speed decreased over complex surfaces, but this finding was not supported by the work of *Thies, Richardson & Ashton-Miller (2005)*. These differences may have arisen because there are no standards for defining complex surfaces in terms of roughness, slope, etc. In contrast to studies of gait, studies investigating gaze during walking have shown a clearer consensus. Compared to smooth surfaces, complex surfaces have been shown to cause eye movements to be increasingly directed to the ground, to lead to increased numbers of fixations, and to require visual information from at least two steps ahead for safe and efficient locomotion (*Matthis & Fajen, 2014*; *Matthis, Yates & Hayhoe, 2018*; *Marigold & Patla, 2007*; *Matthis, Barton & Fajen, 2015*).

Crucially, it is not known whether laboratory simulations accurately represent the surfaces over which we typically walk in everyday life. An alternative, and more ecologically valid, approach to using mixed surface conditions inside the laboratory is to conduct experiments outside. *'t Hart & Einhauser (2012)* assessed gaze for individuals walking outdoors on irregularly placed steps and a smooth road. They reported that their complex surfaces caused individuals to lower both their eyes and head. The eyes lowered more than the head, suggesting that the eyes served more immediate demands when walking. Note, though, that this study only indirectly measured head movements by inferring them from the output of the scene view camera attached to the eye-tracker. Thus, we do not yet have an accurate understanding of how the head affects overall gaze.

Although the results of *'t Hart & Einhauser (2012)* suggest that the head plays an important role in altering gaze when traversing complex surfaces, few other studies have investigated the importance of head movements, independent of eye movements, in contributing to overall gaze. For example, *Matthis & Fajen (2014)* & *Marigold & Patla*

*(2007)* only considered eye movements during walking over complex surfaces. Other studies have inferred head movement from movements of the world camera attached to the eye tracker (*'t Hart & Einhauser, 2012*; *Elloumi, Treuillet & Leconge, 2013*). In the present study we follow *Matthis, Yates & Hayhoe (2018)* using an alternative approach that allowed us to measure head movements independent of the eye tracker whilst walking over complex surfaces. This methodology to calculate gaze has been previously used for tasks other than walking over complex surfaces, for example see (*Fang et al., 2015*; *Land, 1992*). Head movements are particularly important to consider given that weakened musculoskeletal health, including age associated declines, might limit head movement, and this, in turn, could impact gaze. *Tomasi et al. (2016)* assessed head movements, using IMUs, whilst also tracking eye movements. They found that over 40% of gaze movement was due to head movements when walking outdoors. This study did not, though, measure other behaviour changes which are also likely to be important to understanding the relation between locomotion and gaze behaviour, such as speed of locomotion and changes in stride length or timing. Moreover, *Tomasi et al. (2016)* only analysed head yaw (left to right, horizontal movement), whereas head pitch (up and down, vertical movement) is likely to be more important when traversing non-smooth surfaces (*'t Hart & Einhauser, 2012*). Common sense would dictate that movement of the eyes to change vertical gaze orientation are more energy efficient than movement of the head, which requires the activation of more and larger, muscles for the same effect on gaze location. However, to the authors' knowledge, no study has accurately assessed eye angle and head angle when walking over complex surfaces. Thus, it remains unclear how eye and head angle contribute to gaze when walking over different surfaces.

On the basis of the above we believe it is important to independently assess head as well as eye movements to understand how surface complexity influences gaze, and to see how this relates to changes in gait. As an initial step, in this exploratory study, people walked in a straight line on four horizontal surfaces at self-paced speeds. We measured changes in vertical eye angle and head pitch angle, as well as the gait speed of participants. Here, we focus on presenting results for mean values across a trial walk for eye angles, head pitch angles, gaze (combined eye and head pitch) angles and gait speed, as our aim was to compare overall performances across different surface complexities. In future work we aim to conduct more fine-grained analyses of short term, step by step changes in the relation between eye and head pitch angles and gait. For eye and head angle, only vertical change was assessed as horizontal movements are unlikely to be associated with maintaining stability during straight line walking. Thus, in summary, we assessed how eye and head movements independently contribute to gaze, and how this relates to changes in gait speed during locomotion over surfaces of different complexity.

## METHODOLOGY

### Participants

11 healthy adults (7 male, mean ± SD; age = 24.6 ± 3.5 years; height = 173 ± 6.5 cm) were recruited for this exploratory study. Data from 9 more participants was collected but was
not used due to a malfunction of the inertia sensors (both the gyroscopic and accelerometric data recorded for these participants produced extreme values, far exceeding the normal range in all trials). For ease of recording data with the eye tracker, only participants who did not require glasses for everyday walking were selected. No participant had an injury or impairment that affected their gait or vision.

## Data collection

Ethical approval for the study was obtained from the University of Liverpool's Ethics Committee (REF: 1900). Two IMU sensors (Delsys TRIGNO™ IM, Boston, MA, USA) were positioned on the participant. Each sensor consisted of a 3-axis accelerometer, gyroscope and magnetometer, recording at 148 Hz. One IMU was positioned close to the midline of the forehead to calculate head pitch angle using gyroscopic data. The second IMU was positioned above the lateral malleoli on the left shank and was used to calculate gait events. Participants wore an Arrington Research ViewPoint (Scottsdale, Arizona, USA) eye tracker that recorded pupil movement at 60 Hz and a scene camera that recorded the participant's view of the environment that recorded at 30 Hz. Eye angles in the vertical direction were calculated in order to calculate how far ahead on the ground participants were looking as they walked.

## Protocol

The eye-tracker was calibrated prior to each data collection session. Eye movements were calculated based on the dimensions on the screen used in the calibration, see Supplemental Information 1. Participants then walked ten times over four different surfaces so they each completed 40 trials in total. The surfaces comprised an uneven, indoor, and a flat, indoor surface, both in a gait laboratory, and then a paved, outdoor, and a cobbled, outdoor surface, both on the university campus (Figs. 1A–1D). The indoor, flat surface (13.20 m long) consisted of eleven 18 mm thick medium density fibreboards (MDF) panels. The indoor, uneven surface was identical except that each panel had an array of blocks of nine mm thick MDF on top of the base layer to give an uneven surface with a maximum height range of 27 mm. Each panel had the same block design, with blocks spaced to prevent participants from easily targeting footfalls whilst walking. The outdoor, paved surface (16.60 m long) comprised paving stones (60 × 60 cm) whilst the outdoor, cobbled surface (15.70 m long) comprised of setts. All surfaces were long enough to ensure participants could achieve a steady state of walking (Najjar, Iman-Eini & Moeini, 2017).

Participants walked over a wooden obstacle (61 cm wide × 29.5 cm deep × 10 cm high) placed at either the start or end of each surface. This obstacle was intended to increase surface complexity and thus to influence behaviour. However, the location of this obstacle (start versus end) did not show a strong or clear relationship with either gaze angle or speed across any of the four surfaces so this manipulation was not included in the analysis presented here.

On each trial, participants were instructed to begin by looking straight ahead whilst standing still in front of each surface for three seconds, then to walk at a self-determined, comfortable speed along the surface before looking straight ahead whilst standing still at

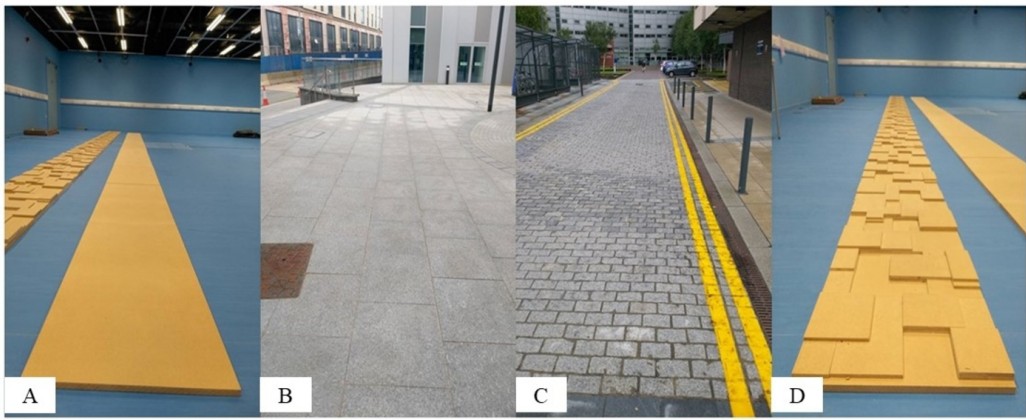

**Figure 1** **Images showing the four surfaces: (A) indoor, flat, (B) outdoor, paved (C) outdoor, cobbled and (D) indoor, uneven.** To estimate surface roughness, we used a clinometer to take 20 measurements of the height change between a pair of points that were 15 cm apart. This was done at 30 cm intervals along each surface. The mean (±SD) height change was 1.8° (±0.5°) for the indoor, flat surface, 1.9° (±0.5°) for the outdoor, paved surface, 2.5° (±1.9°) for the outdoor, cobbled surface and 7.5° (±2.6°) for the indoor, uneven surface.

the end of the surface for three seconds. No instructions were given regarding head or eye movement when walking.

## Data analysis

Mean eye angle and head pitch angle were calculated for each trial of each surface for all participants. For the raw vertical gyroscopic data used to determine head pitch angle, a low pass, 10 Hz fourth-order Butterworth filter was used to reduce noise. The effect of drift was removed using gyroscopic data taken from the period when the participant remained still at the start and end of each trial (following Takeda, Lisco (*Takeda et al., 2014*)). The gyroscopic data (in degrees per second) was then numerically integrated over the trial to give head pitch angle. Supplemental Information 1 describes a check of the accuracy of this method. The vertical eye movements were converted into angular data. A head pitch angle of 0° was defined as the average head orientation at the beginning and end of each trial when the participant remained still whilst looking straight ahead. To avoid the influence of starting and stopping, the walking data was trimmed to remove the first two and last two strides from each trial. Every 1/60s during each trial, the eye angle and head angle were summed and these sums were then averaged across the trial to calculate gaze (combined eye and head pitch) angle for that trial. The relative frequency distribution of eye, head pitch and gaze angles for each surface were also calculated. This measurement follows *Foulsham, Walker & Kingstone (2011)* in calculating the frequency of recorded angles for each surface in bins of 5° relative to zero. In effect this distribution shows the variance of eye and head movement during the trial. Only eye angles that were within the normal range expected based on previous reports (*Lee et al., 2019*) and from our own validation study (see Supplemental Information 1) were included.

Gait speed was calculated using the shank IMU to estimate the shank ankle and then combining this with integrated accelerometery data (following Li, Young *Li et al. (2010)*. As this method has only been tested over smooth surfaces, we checked its accuracy over the most complex surface, the indoor, uneven surface, as detailed in Supplemental Information 1.

Repeated-measures ANOVAs were conducted on the participant's mean eye angles, head pitch angles and gaze angles. The factor of surface had four levels: flat, paved, cobbled and uneven. Correlations were calculated between eye angle and head pitch angle every 1/60s of the trial for all participants. We then conducted t-tests for each surface to compare the mean correlation across participants to a no correlation value (zero). Zero correlation would suggest that there was no relation between eye angle and head pitch angle for that surface. A repeated-measures ANOVA was conducted for the participant's mean gait speed with a factor of surface. Finally, correlations were calculated between mean speed and mean eye angle, mean head pitch angle, and mean gaze, followed by t-tests for each surface to compare the mean correlation across participants to a no correlation value (zero). These correlations were calculated between mean values over a whole trial since speed was calculated across step duration whereas eye, head pitch and gaze angles were calculated every 1/60s.

## RESULTS

### Analysis of the orientation of eye, head pitch and gaze (combined eye and head pitch) angles

Comparisons were made between all four surfaces. Mean ($\pm$SE) eye ($\alpha$/red), head pitch ($\theta$/blue) and gaze (grey) angles ($°$) are shown in Fig. 2. Surface had a significant effect on gaze angle, $F(3, 30) = 28.34$, $\eta_p^2 = 0.81$, $p = 0.003$. Post-hoc Newman Keuls tests ($p < 0.05$) showed gaze to be significantly lower for the indoor, uneven surface compared to the other three surfaces. The contribution to mean gaze angle from head pitch ($\theta$) angle changes were 17% for indoor, flat surfaces; 25% for outdoor, paved surfaces; 35% for outdoor, cobbled surfaces; and 54% for indoor, uneven surfaces. This contribution was calculated as the percentage of the head pitch angle compared to the gaze (combined eye and head pitch) angle taken every 1/60s over the course of every trial and then averaged. The average frequency distribution of eye, head pitch and gaze angles over the trial was calculated for each surface (see Fig. 3). The indoor, uneven surface had a different distribution to the other three surfaces. These other surfaces all had peak head pitch angles close to zero, whereas the indoor uneven surface showed a greater range of head pitch angles. For this surface, head pitch angle was often lowered, with a similar range distribution to that for eye angle. Gaze (combined eye and head pitch) angle showed a similar distribution to eye angle for all but the indoor, uneven surface where it was generally lower.

A one sample t test showed that the correlation between eye and head pitch angle for the indoor, flat (M $\pm$ SD = +0.13 $\pm$ 0.17), and outdoor, paved surface, (+0.24 $\pm$ 0.15) were significantly greater than zero (t (10) = 2.46, $p = 0.034$ and t(10) = 5.63, $p < 0.01$ respectively). These correlations, albeit weak, suggest that eye and head movements are

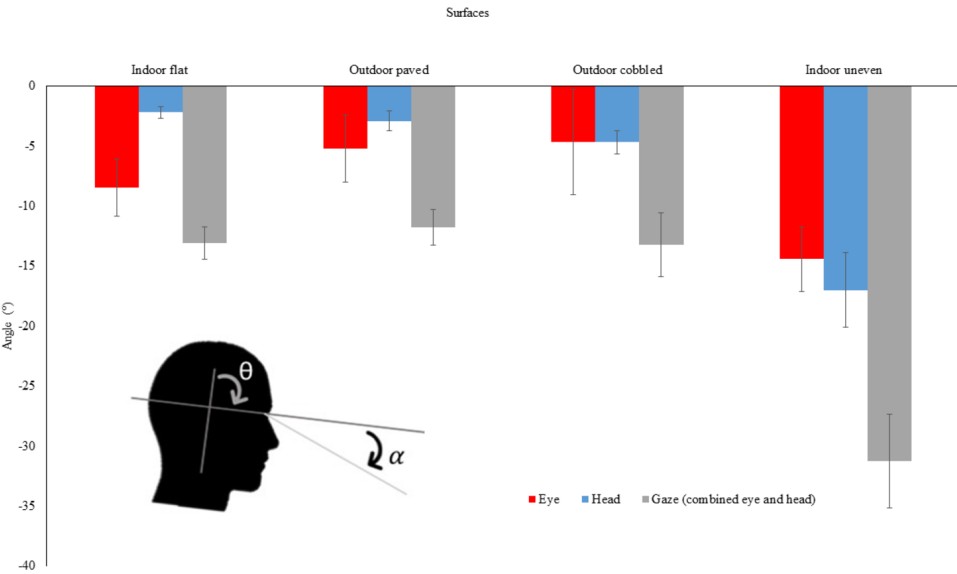

**Figure 2** Mean (±SE) eye, head pitch and gaze (combined eye and head pitch) angles (°) for the four surfaces tested: indoor, flat; outdoor, paved; outdoor, cobbled; and indoor, uneven. The inset shows how eye (α/red) and head pitch (θ/blue) angles were measured. Mean gaze (combined eye and head pitch) angle is the mean value of the sum of eye angle and head pitch angle calculated every 1/60s (and not the sum of the mean eye angle and mean head pitch angle).

co-ordinated when walking over these surfaces. The correlations for the indoor, uneven ($+0.01 \pm 0.17$; t(10) = 0.26, $p = 0.801$) and outdoor, cobbled ($+0.15 \pm 0.23$; t(10) = 2.12, *p=0.060)* surfaces were not significantly different to zero. As all four correlations were all relatively low, this suggests that eye angle and head pitch angle both contribute distinct information about gaze angle.

## Gait speed analyses

Speeds were significantly different across surfaces, $F(3, 30) = 38.40$, $\eta_p^2 = 0.89$ $p < 0.001$, as shown in Fig. 4. A post-hoc Newman Keuls test ($p < 0.05$) showed participants walked more slowly on the indoor, uneven surface (M ± SE = 1.19 metres/second ± 0.05*)* than the indoor, flat ($1.35 \pm 0.04$), outdoor, paved ($1.43 \pm 0.04$) and outdoor, cobbled ($1.44 \pm 0.04$) surfaces.

Correlations of speed were calculated between mean speed and mean eye, head pitch and gaze angle over the trial. One sample *t*-tests revealed that no correlations with speed were significantly different from zero see (Table 1).

We estimated how long it would take participants to walk to the location that they were fixating for each surface. To do this, we used the average participant eye height and their mean gaze (combined eye and head pitch) angle to calculate the mean distance that participants were looking ahead for each surface. We then divided this distance by the average participant gait speed for that surface to estimate how long it would take for participants to walk to their fixation location (see Table 2). This was shortest for the indoor uneven surface. Similarly, using the average step length for each surface, we calculated how
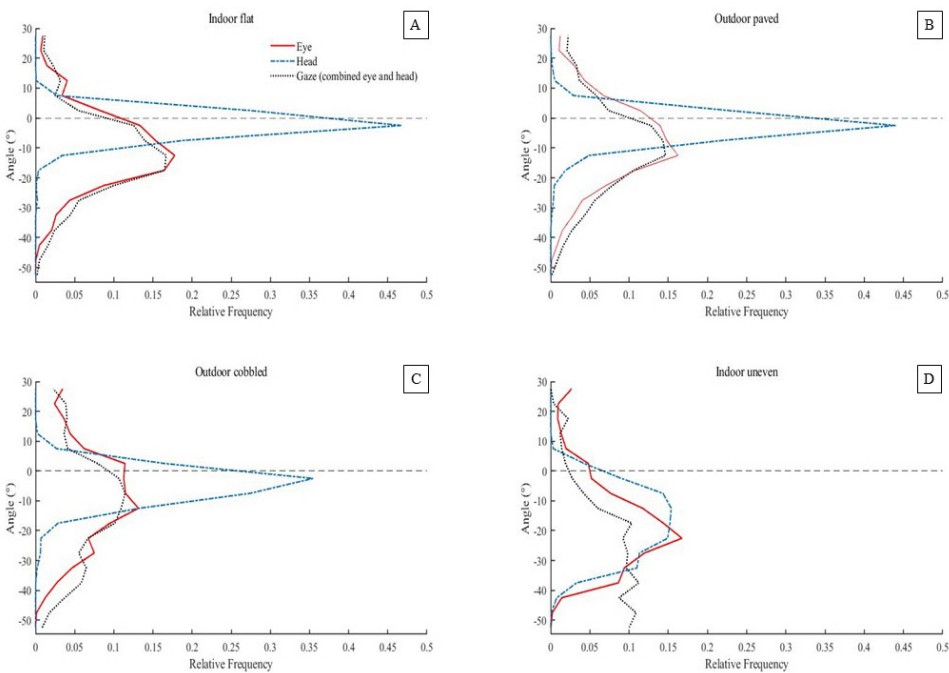

**Figure 3** **Relative frequency distributions of eye, head pitch and gaze (combined eye and head pitch) angles (°), within a trial for the (A) indoor, flat, (B) outdoor, paved, (C) outdoor, cobbled and (D) indoor, uneven surfaces.** Negative angles correspond to lowering of the eyes and head toward the ground. An angle of zero (indicated by the black dashed line) represents the mean angle as the participant looked ahead at the start and end of each trial.

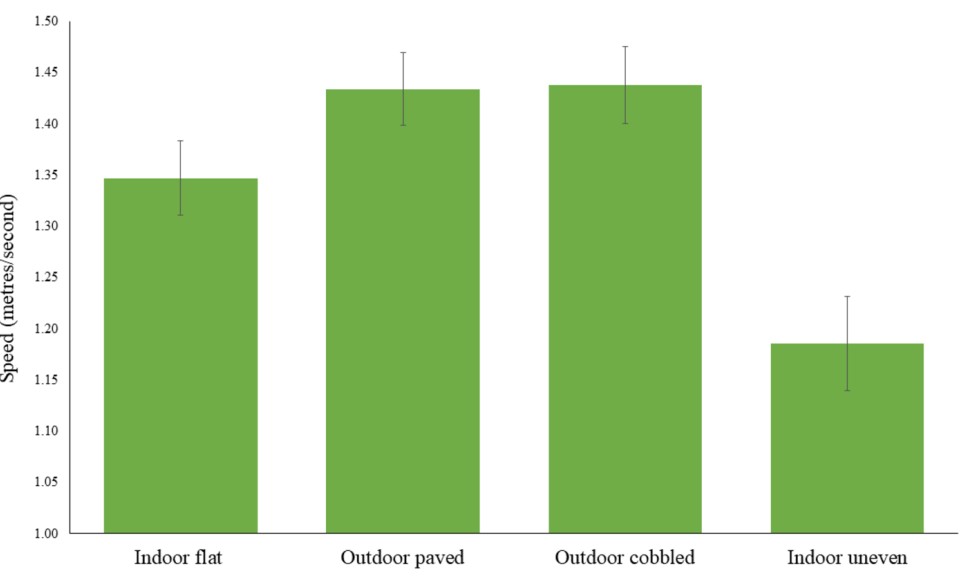

**Figure 4** **Mean (±SE) gait speed (metres/second) for the four surfaces (indoor, flat, outdoor, paved, outdoor, cobbled and indoor, uneven).**

**Table 1  Correlations between mean gait speed and mean eye, head pitch and gaze angles (°).**

|  |  | Indoor, flat | Outdoor, paved | Outdoor, cobbled | Indoor, uneven |
|---|---|---|---|---|---|
| | **Mean (±SD)** | **+0.01 (±0.29)** | **−0.14 (±0.37)** | **−0.04 (±0.39)** | **+0.09 (±0.28)** |
| Eye angle (°) | *t* value | 0.08 | −1.21 | −0.32 | 1.07 |
| | *p value* | *0.936* | *0.255* | *0.755* | *0.311* |
| | **Mean (±SD)** | **−0.08 (±0.37)** | **+0.09 (±0.36)** | **−0.06 (±0.41)** | **−0.11 (±0.35)** |
| Head pitch angle (°) | *t* value | −0.69 | 0.83 | −0.52 | −1.07 |
| | *p value* | *0.508* | *0.428* | *0.613* | *0.310* |
| | **Mean (±SD)** | **+0.03 (±0.33)** | **+0.06 (±0.35)** | **+0.06 (±0.41)** | **+0.07 (±0.32)** |
| Gaze angle (°) | *t* value | 0.28 | 0.54 | 0.46 | 0.67 |
| | *p value* | *0.789* | *0.601* | *0.654* | *0.516* |

**Table 2  Mean (±SD) time (seconds) and mean number of steps to reach the location that participants were looking ahead to.**

|  | Indoor, flat | Outdoor, paved | Outdoor, cobbled | Indoor, uneven |
|---|---|---|---|---|
| Look ahead time (sec) (±SD) | 6.37 (±0.12) | 7.88 (±0.12) | 6.82 (±0.12) | 2.23 (±0.15) |
| Look ahead step number (±SD) | 6.12 (±0.22) | 7.75 (±0.29) | 6.66 (±0.25) | 2.00 (±0.07) |

many steps people looked ahead. People looked fewer steps ahead on the indoor uneven surfaces, see Table 2.

## DISCUSSION

The aim of this exploratory study was to understand how eye angle and head pitch angle contribute to gaze behaviour and how this alters with gait speed when walking over surfaces of different complexity. When traversing the most complex surface (indoor, uneven; mean height change = 7.46°, see Fig. 1), participants significantly lowered their gaze (combined eye and head pitch) angle and reduced their gait speed. Head pitch angle was lowered towards the ground for a greater duration of the trial over this surface (as shown by the relative frequency distribution, see Fig. 3), and a greater proportion of gaze angle was attributed to head pitch angle than for any other surface (54%). Our results suggest that more complex surfaces require greater visual information to traverse, with a stronger contribution to overall gaze angle being made by head pitch angle in such circumstances.

The results in our study are consistent with previous research in showing that complex surfaces exert increased visual demands (*Matthis, Yates & Hayhoe, 2018*; *Marigold & Patla, 2007*; *'t Hart & Einhauser, 2012*) as it becomes harder to maintain stability. Using mean values of gaze (combined eye and head pitch) angle and speed, we showed that participants walking over the indoor, uneven surface looked just two steps ahead (see Table 1). This finding is in line with that previously reported when walking on inconsistently spaced foot holds (*Matthis & Fajen, 2014*; *Matthis, Barton & Fajen, 2017*). Further research is required to test how different characteristics of irregular surfaces (slope, unevenness, appearance, texture, etc.) influence eye and head pitch behaviour. The present study only measured

surfaces by changes in their mean height. An important future goal will be to characterise surfaces using comprehensive, objective and replicable measures.

A relatively novel aspect of the current study was analysing eye and head pitch angle independently when walking over different surfaces. Our results found no strong relation between eye and head pitch angle (note, though, that our analyses could not detect short-term correlations). Only two surfaces (indoor, flat and outdoor, paved) produced a significant correlation between eye and head pitch angles and these correlations were weak. For these simpler surfaces there was some evidence that eye and head movements were co-ordinated. This might reflect participants spending more time gazing around the scene rather than having to fixate near to their upcoming foot placements on these less challenging surfaces. The relative frequency plots (see Fig. 3) showed differences between eye and head pitch angles. The eyes were typically lowered more than the head except when walking over the most complex indoor, uneven surface. This suggests that the energetically costly movement of the head to shift gaze is only implemented when necessary, i.e., when surfaces are more complex to traverse, compromising stability. This supports findings from *'t Hart & Einhauser (2012)*, showing eye movements are usually greater than head movements. Furthermore, these results strengthens the rationale of *Tomasi et al. (2016)* for calculating gaze from *both* eye and head movements. The lack of contribution from the head to overall gaze when walking over smooth surfaces may suggest that our peripheral vision is sufficient in these settings. Indeed, peripheral vision has been shown to be sufficient even when traversing obstacles (*Graci, Elliott & Buckley, 2010*). Further research is therefore required to determine how complex surfaces must be in order to elicit lowering of the head.

In our study, changes in eye angle, head pitch angle and gait speed were assessed from mean values across the entire length of the surface traversed on a given trial. We found significant differences between surfaces using this approach (see Fig. 2), and we believe that this summary measure provides a convenient and meaningful summary of gaze behaviour over different surfaces. Surface lengths changed slightly between surfaces, but given that we excluded data from the start and end of the surface, differences of surface complexity are likely to be the main cause of behavioural change.

A more detailed approach to determine gaze behaviour can come from time series data, for example as used by *Matthis, Yates & Hayhoe (2018)*. Supplemental Information 1 shows an example of time series data from our study, plotting raw gaze angle for ten trials of one participant walking over the outdoor, cobbled and the indoor, uneven surfaces. For this participant, gaze was consistently lower for the indoor, uneven surface compared to the outdoor, paved surface, whilst overall gaze angles were generally lower.

## CONCLUSIONS

In summary, we found gaze and gait behaviour to be most affected when participants walked on a complex, uneven surface. In this situation both head and eye movements played a substantial role in determining gaze angle, supporting the argument (*Tomasi et al., 2016*) that we should not assess gaze solely by considering eye movements. This research

should act as a foundation for future work to tease apart what surface characteristics drive behavioural changes in gaze and gait when we walk over the types of surfaces that we commonly encounter in our everyday lives (e.g., slopes, cobbles, steps).

## ACKNOWLEDGEMENTS

The authors wish to thank Jenni Delight and Alexandra Allen for their assistance.

### Funding

This research was supported by a grant from the Economic and Social Research Council to the first author [ES/J500094/1]. The funders had no role in study design, data collection and analysis, decision to publish, or preparation of the manuscript.

### Grant Disclosures

The following grant information was disclosed by the authors:
Economic and Social Research Council: ES/J500094/1.

### Competing Interests

The authors declare there are no competing interests.

### Author Contributions

- Nicholas D.A. Thomas conceived and designed the experiments, performed the experiments, analyzed the data, prepared figures and/or tables, and approved the final draft.
- James D. Gardiner, Robin H. Crompton and Rebecca Lawson conceived and designed the experiments, authored or reviewed drafts of the paper, and approved the final draft.

### Human Ethics

The following information was supplied relating to ethical approvals (i.e., approving body and any reference numbers):

Ethical approval for the study was obtained from the University of Liverpool's Ethics Committee (REF: 1900).

### Data Availability

Data is available at Figshare:

1. Thomas, Nicholas (2019): Look_Out_v2_Experiment _Data.zip. figshare. Dataset. DOI 10.6084/m9.figshare.9873278.v1.

2. Thomas, Nicholas (2019): analyse_results.m. figshare. Dataset. DOI 10.6084/m9.figshare.11336846.v1.

3. Thomas, Nicholas (2020): Excel_files_eye_head_time-frame_data. figshare. Dataset. DOI 10.6084/m9.figshare.11536413.v1.

## Supplemental Information

Supplemental information for this article can be found online at http://dx.doi.org/10.7717/peerj.8838#supplemental-information.

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
