# Peer review of "Look out: an exploratory study assessing how gaze (eye angle and head angle) and gait speed are influenced by surface complexity"

_PeerJ, doi:10.7717/peerj.8838_

## Round 0.1 · original submission · Major Revisions

Please revise your paper based on the comments of two reviewers.

Reviewer 1 ·

Basic reporting

This paper explores the relationship between gaze angle, gait speed and the complexity of surfaces by simultaneously measuring the orientation of the eyes with an eye tracker and the orientation of the head with an IMU.

The paper is well written and the design of the study is adequate for the aim of the study. In addition, I agree with the authors that when studying gaze in natural conditions head movements should be considered. Practically, this is not trivial and it can be challenging to measure head and eye movements simultaneously. Furthermore, it is good to see that the IMU was compared to a MOCAP system, this demonstrates that the authors are well aware of potential issues in their set-up.

However, I was slightly disappointed after reading the results section. I feel that the authors oversimplify their dataset by only analyzing the average head angle and average eye angle during the different surface conditions. This is a pity considering the rich dataset the authors have at their disposal. In addition, due to the limited analyses, I feel that the conclusions drawn in this paper are not yet supported by the data.

Experimental design

The design of the study is adequate, but I do have serious concerns about the analyses. Please find specific comments below.

Validity of the findings

Due to the limited analyses, I feel that the conclusions drawn in this paper are not yet supported by the data. Please find specific comments below.

Additional comments

1. The main reason for this study appears to be that previously most studies only analyzed the eye movements of the participants and neglected the head movements. I certainly agree that this can be problematic, especially in natural conditions and in relation to surface complexity. However, in this study, the eye movements and head movements are completely separated. And although the authors mention in the introduction that: “[while] gaze is often used by researchers to refer only to eye movements, here we define gaze as the combined position of the eye and the head”, this appear to be completely overlooked in the results of this study. The only reference to gaze in the results section is in the first paragraph, in which is referred to Figure 2 and Figure 3. However, in both figures, the gaze is not shown, only the head pitch and elevation angles of the eye movements are plotted. It is difficult to interpret both figures without this information.

For example, the plot in Figure 3 of the relative frequency distribution for the “indoor uneven” surface could be the result of participants keeping their gaze steady and effectively cancelling the head movements with their eye movements (i.e. head about 20 deg down from the average position, while counter rotating about 20 deg up with the eyes to keep gaze stable using the vestibular ocular reflex, thus a negative correlation between eye and head), or by vertically scanning the environment (i.e. a large eye movement of the head accompanied by a large eye movement in the same direction, thus a positive correlation between eye and head), of by uncorrelated movement of the eye and head.

In addition, the results of the ANOVA reported in the first paragraph of the results (lines 200-203) do suggest that the gaze was calculated, but further inspection reveals that the main effect of the ANOVA and the subsequent reported values of “gaze” are only the mean of the average head and eye angles. And as I just pointed out that does not necessarily reflect the behavior of the participant. The same appears to be the case for the contribution to gaze from head pitch in lines 203-204.

In sum, instead of inferring gaze based on average head and eye angles,
the authors should first calculate the gaze orientation based on the individual raw (or filtered) eye and head traces, before averaging the angles for different surface. Which is in fact exactly what Tomassi et al, 2016 did (the authors do suggest that they followed their approach in line 67, but they actually do not).

2. Furthermore, I feel that only presenting the averaged angles is an oversimplification of the behavior of the participants. I assume that they did not keep a fixed eye and head orientation during the trials. It would be beneficial to analyze the data further, for instance looking into the variance of eye and head movements during trials. I can imagine that they explored the complicated surfaces more, probably reflected in more saccades and/or larger variance in gaze orientation (and probably also in head/eye angles). In addition, I would like to see some data traces of eye, head and gaze angles during trials on the different surfaces (representative trials would be a good start, trying to visualize the data of all participants even better).
In addition, it might be interesting to have a look at gaze projection on the ground, i.e. how far ahead they directed their gaze, especially in combination with the speed of gait.

3. In the abstract it is stated that: “Eye and head pitch angles did not correlate across any surface, thus can be considered independent measures.” However, only the average head and gaze angles of the participants were correlated. I assume that the participants actually did coordinate their eye and head movements during the trials, at least to some extent, to keep a stable image on the retina. You made the data of the head and eye movements independent by first averaging them, so it is not a surprise that your analysis does not show a dependence. The same goes for the statements in the abstract that: “[it] provides the novel finding that head movements provide important independent contributions to gaze location.” You cannot conclude this based on your current analyses.

4. Data from 9 of the 20 participants was excluded. Although I know that these kind of set-up can be technically challenging and sometimes eye trackers have troubles tracking certain eyes etc, excluding almost half of the participants is too much to justify it only based on the vague reason of technical errors. Please elaborate on the reasons why the measurements failed for these participants. In addition, I noticed that there is actually data on all surfaces for these participants in the dataset, was all data useless, or did you exclude participants if the data quality was insufficient on one or more surface (and if so, based on what kind of criteria).

5. I tried to have a look at the raw data. However, without additional information, I could not explore the data myself. Please provide preprocessed data, e.g. head pitch and eye orientation over time with time stamps that allow to synchronize the data.

6. In the introduction it is stated that “For example, Matthis and Fajen [1] & Marigold and Patla [9] only considered eye movement during walking over complex surfaces.”. And although most studies did not specifically analyze the kinematics of the head and eyes simultaneously, several studies actually did measure the 3d gaze orientation, thus incorporating simultaneous measurements of head movements and eye movements. See for instance reference [3]. Mathis et al., 2018, in which they looked at the 2D projected position of gaze on the ground. In addition, there is more literature about eye-head coordination in other tasks. Please discuss these studies in the introduction.

7. In Figure 3, the distribution of head and eye angles are plotted for the different surfaces. I noted that, although the relative frequency is low, that there are measured eye angles of >30 degrees. Considering the normal range of eye movements (see e.g. Lee et al. 2019, EYE, Differences in eye movement range based on age and gaze direction), I wonder whether these might be artefacts rather than actual eye movements.

Furthermore, please change the label on the y-axis. You do not plot gaze, but only the orientation of the eye and head.

8. Supplementals: Please provide more detail about your set-up. What kind of IMU’s did you use and what kind of MOCAP system? And please provide additional information about the marker placement, i.e. number of markers and placement of markers, and how you calculated head pitch and speed of gait from that.

Reviewer 2 ·

Basic reporting

It is clear and well written with sufficient background and a good structure.

I can confirm that raw data is shared on fig share.

Experimental design

This is original research which addresses a meaningful question. It is performed to a rigorous standard.

Methods-wise, I think more information could be given about how the gaze angles are calculated as at present it would be hard to replicate this. The eye tracker presumably gives eye-in-head data and/or the point-of-regard in the scene camera. I'm not sure whether this is what is analysed or if the authors have head position or just work out the offset relative to gravity.
It is also not clear how the "percentage contribution of eye or head" to gaze is calculated.
It is also not clear how the eyetracker was calibrated.

Validity of the findings

Everything seems robust here.

I didn't really understand why eye and head were correlated in the way that they were. It seems like we would expect an anti correlation (because if the head points down the eyes would have to point up maintain fixation on the same point.

---

## Round 0.2 · Minor Revisions

Thanks for re-submitting to PeerJ. Please revise your paper according to the final review comments.

Reviewer 1 ·

Basic reporting

The authors have comprehensively responded to my comments and have sufficiently addressed all my major concerns. I am particularly pleased to see the authors have added the gaze data to the manuscript, that they have changed their conclusions accordingly and that they have added the scripts and additional data to facilitate exploration of their data by others.

There are a few minor issues that needs attention, but they should be quite easy to fix.

General:
- Please be consequent in terminology about gaze/head/eye direction. Now different words are used which do not always have the same meaning in this context: orientation, position (which is not the same as orientation in this case), angle , pitch, and direction.
- I appreciate that the authors added the data on the gaze projection on the ground with the speed of gait in the supplementary materials. I think that it would be a valuable addition to the main manuscript and would therefore encourage the authors to move the table from the supplemental materials to the manuscript.

Title:
- I prefer “speed of gait” instead of “gait speed”. Please change throughout the manuscript. I suspect that you made this choice because of the word count of the abstract, but I think that this can be fixed easily by removing one sentence in the significance section of the abstract.

Abstract:
- Lines 29-31: wording is a bit awkward. It is now insufficiently stated that previously only the eye movements were measured, but that head movements contribute to gaze as well and that therefore it is important to measure both simultaneously. In your introduction you already mention that: “it is important to independently assess head as well as eye movements to understand how surface complexity influences gaze, and to see how this relates to changes in gait.” Please rephrase the background section of the abstract.
- Research question: I think that the wording in sentence I mentioned above is stronger. Please consider rephrasing the research question
- Results: I would change the order, by first describing the results of surface complexity on gaze and then indicate that head movements play an important part in the changes in gaze.
- Significance: Considering the limited word count I would suggest removing the last sentence.

Introduction:
- Lines 66-67: “Lastly” to “head movements.” Your definition of gaze is not clear enough. Gaze is the combination, or result’ of head and eye movements. Gaze is effectively the orientation of the eye in a world reference frame, in this case calculated based on eye-in-head data of the eye tracker and head-in-space of the IMU.

Methods
- I appreciate the addition of the reason why you had to exclude the data of 9 participants.
- Lines 147-148: “Vertical pupil” to “they walked”. This is rather vague. I assume that you used the eye-in-head data in degrees. And you need to use the gaze data to calculate how far ahead on the ground participants were looking, i.e. combination of eye and head data. For me, vertical pupil movement refers to the uncalibrated data of the movement of the pupil in pixels in the video data.
- Repeated measures ANOVA only on eye and head data? Why not on the gaze data as well. That will provide the answer to your main research question, i.e. to understand how surface complexity influences gaze. Edit: Based on the results it appears that you actually performed the ANOVA on the gaze data. In that case, add it to the methods.

Results:
- Line 210. This heading should be rephrased for clarity. For example something like this: Analyses of the orientation of the eye, head and gaze
- The positive correlations between eye and head angles for the indoor flat and outdoor paved surface are a bit surprising. In my opinion, it indicates a coordinated scanning pattern (i.e. head lowered accompanied by lowering of eyes). I think it would be good to explain this in the result section and maybe elaborate a bit more on this in the discussion section.
- Lines 240-249: Why not calculate the correlation between gaze and speed of gait? Again, that is in line with your main goal.

Discussion:
- Lines 252-253: Here the aim is clearly stated, but it differs slightly from what is stated in the abstract and introduction.
- Lines 263-265: I think this is a relevant addition to your results and therefore it belongs in the results section, not in the supplementary materials.
- Lines 274-275: I think that the statement that “eye and head pitch angle measures were not
predictive of one another” is a bit too strong considering the analyses you did. I suspect that in reality, periods of positive correlations (e.g. making a gaze saccade to the next point of interest) are alternated with periods of negative correlations (e.g. keeping gaze stable at point of interest with the VOR to counteract the movement of the head due to walking). You only looked at the mean correlation. And therefore, perfect alternation of positive and negative correlation periods, and thus perfect coordination of eyes and head, would in your analysis result in an average correlation of zero.

Experimental design

No additional comments.

Validity of the findings

No additional comments.

Additional comments

No additional comments.

Reviewer 2 ·

Basic reporting

no comment.

Experimental design

no comment.

Validity of the findings

no comment.

Additional comments

The authors have addressed the minor analysis/method points I raised and the manuscript is now acceptable.

---

## Round 0.3 · accepted · Accept

Thanks for your revision.